# Review about Powerful Combinations of Advanced and Hyphenated Sample Introduction Techniques with Inductively Coupled Plasma-Mass Spectrometry (ICP-MS) for Elucidating Trace Element Species in Pathologic Conditions on a Molecular Level

**DOI:** 10.3390/ijms23116109

**Published:** 2022-05-29

**Authors:** Bernhard Michalke

**Affiliations:** Research Unit Analytical BioGeoChemistry, Helmholtz Zentrum München—German Research Center for Environmental Health GmbH, 85764 Neuherberg, Germany; bernhard.michalke@helmholtz-muenchen.de

**Keywords:** LA-ICP-MS, spatial element mapping, elemental tagging, iron redox speciation, copper speciation, single cell ICP-MS (SC-ICP-MS)

## Abstract

Element analysis in clinical or biological samples is important due to the essential role in clinical diagnostics, drug development, and drug-effect monitoring. Particularly, the specific forms of element binding, actual redox state, or their spatial distribution in tissue or in single cells are of interest in medical research. This review summarized exciting combinations of sophisticated sample delivery systems hyphenated to inductively coupled plasma-mass spectrometry (ICP-MS), enabling a broadening of information beyond the well-established outstanding detection capability. Deeper insights into pathological disease processes or intracellular distribution of active substances were provided, enabling a better understanding of biological processes and their dynamics. Examples were presented from spatial elemental mapping in tissue, cells, or spheroids, also considering elemental tagging. The use of natural or artificial tags for drug monitoring was shown. In the context of oxidative stress and ferroptosis iron, redox speciation gained importance. Quantification methods for Fe^2+^, Fe^3+^, and ferritin-bound iron were introduced. In Wilson’s disease, free and exchangeable copper play decisive roles; the respective paragraph provided information about hyphenated Cu speciation techniques, which provide their fast and reliable quantification. Finally, single cell ICP-MS provides highly valuable information on cell-to-cell variance, insights into uptake of metal-containing drugs, and their accumulation and release on the single-cell level.

## 1. Introduction

The analysis of chemical elements in clinical or biological samples gained eminent attention, due to their essential roles in life science, clinical diagnostics, and drug development and drug-effect monitoring [1,2,3,4,5]. Particularly in body fluids, such as serum or cerebrospinal fluid, as well as in tissue samples, the content of chemical elements, the specific element binding forms (element species), their redox state under specific health or pathologic conditions, as well as their spatial distribution in tissue of pathological interest, or even in single cells, are the focus of research today. Elements, even in trace amounts, have a profound influence on the regulation of biochemical processes in humans. On the one hand, around one third of the enzymes are metalloenzymes, which have to be supplied at their prosthetic groups with a sufficient amount of the respective metal in order to be able to fulfill their biochemical function. On the other hand, the cellular redox balance—decisively influenced by the ratio of different valence states of some transition metals (Fe, Mn, Cu) [6,7,8,9,10]—can be significantly shifted towards the oxidative stress (OS) condition. Lipid peroxidation and cellular decline may follow OS, resulting in disease onset or progression. Another matter of interest is the monitoring of metal drugs, such as platinum complexes, in cancer therapy. Their distribution or conjugation with proteins may reduce their effectiveness [11].

For more than two decades, ICP-MS demonstrated its superior capability in measuring the elements of biological or clinical origin at the lowest concentrations with high linearity, from <1 ng/L to the high mg/L range. Owing to its outstanding sensitivity, its application in life sciences rapidly grew and, nowadays, is well-established [12]. Furthermore, technical developments and improvements in ICP-MS technology, such as dynamic reaction cell technology combined with multi-quad-MS, now allow for measurements of interfered, previously difficult to determine, isotopes, with high-precision and accuracy. In parallel to this improvement in ICP-MS-instruments, different “advanced and specific sample introduction systems” were developed for special analytical purposes, which are now even commercially available, and together with sensitive ICP-MS detection, drastically enhance the information density of the (clinical) samples in question. The first combinations published used high-performance liquid chromatography (HPLC) systems as a sample introduction; alternatively, ICP-MS was used as element/isotope-specific chromatographic detector, capable of providing several element-specific chromatograms at the same time from one sample. Depending on the choice of chromatography type (e.g., size-exclusion chromatography (SEC), ion-exchange chromatography (IEC)), element species information was extracted about the distribution of elements to big or small proteins or low-molecular-weight compounds (by SEC) [13,14], or about ion-exchange properties of metal-carrying proteins, and particularly the valence state of metals in cellular extracts or body fluids (by IEC) [15,16,17,18]. This promising combination of chromatographic element species separation and element selective detection was adapted and applied to numerous scientifically and clinically relevant problems, and in the following years, further extended to combinations of multi-dimensional separations (e.g., HPLC-capillary electrophoresis (CE)) with coupled ICP-MS, and/or electrospray ionization tandem mass spectrometry (ESI-MS/MS) detection. These combined methods provided orthogonal and/or molecular identification of metalloid compounds. For spatial element mapping in tissues, the laser ablation in front of ICP-MS was the combination of choice. However, all such innovative and powerful analytical approaches were mainly developed and applied on comparatively small sample sizes in analytical research institutes. On occasion within analytical–clinical cooperation, the techniques found their way, typically, into limited clinical application, but never into routine clinical practice. Two main reasons may be identified for this unfortunate circumstance. Firstly, these methods, and their comprehensive and very powerful potential for information gain, are often not familiar to medical researchers. The development and application of these combined methods is usually found in analytical chemical laboratories, published as “proof of principle” in analytical chemical journals, using well-established chemical–physical terminology in order, amongst other things, to show the analytical quality and rigidity of the new method. Scientists trained in medicine or molecular biology, however, tend to obtain professional information from “their field”, namely biochemical, cell-biological, medical, or molecular biological journals. Journals that seek an intersection of analytical chemistry and the biomedical disciplines are rare and, unfortunately, are often not considered as the first choice by either side, either for publication submission or for reading; typically the top journals of each separate field are preferred. The second reason that these hyphenated techniques are not used in clinical routine is certainly the high price per analysis, and the increased timespan required per analysis, which is a substantial limitation for high-throughput approaches.

The paper introduces powerful combinations of specific sample introduction systems hyphenated to ICP-MS for diagnostic, monitoring, and clinical research applications. The manuscript is intended as a brief introduction to those powerful instrumental capabilities, in order to promote the consideration of these techniques in clinical research. This may lead to benefits for the analytical chemical research, through access to relevant clinical samples, as well as acquiring hot topic foci for new instrumental method developments aimed at improvement in medical research. For the biological–medical side, the advantages will be achieved by gaining new and deeper insights into pathological processes at the molecular level, using powerful tools for diagnosis by employing these methods.

For presenting some of these powerful ICP-MS related methods, the paper highlights the fields of spatial elemental mapping, e.g., in brain tissue, tumor cells, or spheroids, and considers also elemental tagging. The latter can work with natural tags, but, nowadays, it is also extended to artificial tags for drug monitoring. Other important areas are redox speciation, for example in the context of oxidative stress and ferroptosis. New and fast methods for the quantification of Fe^2+^, Fe^3+^, and ferritin-bound iron are reported, which now enable the quantitative determination of Fe^2+^/Fe^3+^ ratio, which is the measure for evaluating the iron-related cellular redox status and risk for oxidative stress. In connection with Wilson’s disease, free copper ions and exchangeable copper play a decisive role and should be quantified aside from tightly ceruloplasmin-bound copper. The respective chapter presents information about novel methods based on hyphenated Cu speciation techniques, which are faster and more reliable with respect to quantification than the presently used “difference method”. Finally, new analytical approaches were successful in measuring single cell content of elements. Analysis of individual cells provides highly valuable information on cell-to-cell variance, especially within an isogenic cell population. In contrast to bulk analysis, the information on metal distribution among the cell population gives further insights into uptake of e.g., metal-containing drugs, and their accumulation and release on the single-cell level.

## 2. Methods

The presented manuscript highlighted these exciting analytical advances and it illustrated them with examples from the literature. Example references were identified through PubMed and Google Search using the section-wise listed search items: for Section 3: laser ablation (LA), LA-ICP-MS, spatial element mapping; for Section 4: artificial elemental tags, elemental tagging, natural elemental tags; for Section 5: iron redox speciation, Fe^2+^, Fe^3+^, ferroptosis; for Section 6: Wilson’s disease, copper speciation, free copper; for Section 7: single cell ICP-MS.

## 3. Laser Ablation Inductively Coupled Plasma Mass Spectrometry (LA-ICP-MS)—Spatial Element Mapping

Among the developed bioimaging techniques, laser ablation (LA) in combination with ICP-MS gained a leading role in the field. It allows elemental mapping typically in flat samples, such as tissue sections of various biological and different organ origin, with sufficiently high spatial resolution power; spatial resolution down to the sub-micrometer range with outstanding sensitivity (low μg g^−1^) are reported [19,20]. The monitoring of metal drug distribution in tissue or cells, or metal drug association with proteins, is one of the most prominent fields for LA-ICP-MS application. Theiner et al. [21] report on bioimaging studies in the metallodrug research field. The study is still mainly focused on analytical method development. They apply LA-ICP-MS imaging in multicellular tumor spheroids, as a novel tool in the pre-clinical development of metal-based anti-cancer drugs. Notably, drug tumor penetration is a pre-clinically relevant key question, which has to be addressed for data-based assessment of the potency and effectiveness of metal drugs. LA-ICP-MS offers perfect insights into metal drug distribution inside the neoplasm tissue. Theiner et al.’s methodical achievements show that the sensitivity and a spatial resolution of about 10 μm allows for the localizing of platinum hotspots at the periphery of the spheroids, as well as at the necrotic core. The LA-ICP-MS-generated distribution maps of platinum in HCT116 and CH1/PA-1 tumor spheroids, after treatment with the respective platinum compounds, were correlated with the grey-scaled photo-images, taken prior to the ablation of the samples. This is illustrated in Figure 1. The platinum distribution in the tumor spheroid sections is heterogeneous for both cell lines, with a strong platinum accumulation in distinct areas, as highlighted by red marked dots. 

In a follow-up study, Theiner et al. modify their system and coupled a fast-scanning time-of-flight mass-spectrometer (TOFMS) to their laser ablation system [22]. To obtain not only platinum, but even multi-element information, ICP-MS instruments equipped with time-of-flight mass analyzers are the instruments of choice, as they provide the required simultaneous analysis and fast data read-out. The mapping of Pt drug distribution, in parallel to distribution patterns of the elements Mg, P, S, Ca, Fe, Cu, and Zn—standing for important biological key functions—is an important factor to assess the therapeutic response to chemotherapy. Therefore, in this study, the platinum accumulation, together with the distribution patterns of Mg, P, S, Ca, Fe, Cu, and Zn are studied in germ cell tumors of patients undergoing platinum-based chemotherapy by LA-ICP- TOFMS. As a result, heterogeneous elemental distribution patterns are obtained, which relate to histological analysis after hematoxylin–eosin staining. Necrotic areas show the highest platinum values, as well as high levels of magnesium, sulfur, and calcium, but no visible accumulation of iron. As an example, the overlay of platinum and iron intensity maps of one of the patients reveals a reverse correlation of these two elements in the tumor section, as shown in Figure 2.

So far, LA-ICP-MS is used mainly as a research tool in several bioimaging studies, but not yet in routine applications. However, for its application as a clinical or diagnostic tool, accelerated throughput and high-spatial resolution are paramount. For the improvement of spatial resolution, a small laser spot size is required. Unfortunately, this, in turn, negatively affects detection sensitivity, since a severely decreased number of detectable atoms is ablated [23]. In this context, the combination of LA-ICP-MS with immunohistochemistry promises to become a solution. 

The first immunoassay and detection of metal-labelled antibodies with ICP-MS is introduced by the research group of Scott Tanner [24]. This approach is further developed to a three-fold multiplex by Giesen et al. [25]. These authors match three different biomarkers for breast cancer. In their 2014 paper, Giesen et al. reach subcellular resolution with 1 μm for tissue samples in a multiplex, immune-imaging set-up, using polymer-tagged antibodies and LA-ICP-TOFMS. Element clustering is a comparatively novel way to further increase the sensitivity of above-mentioned approaches. By conjugating with antibodies, even clusters of atoms of selected elements, which additionally show high-sensitivity in ICP-MS detection, demonstrate a drastic increase in signal intensity. As an example, Cruz-Alonso et al. employ antibodies bioconjugated with gold nanoclusters, and, by that way, provide a high-amplification factor compared to DOTA-based labels [26]. A variation of this promising approach is applied by Neumann et al. These authors report about the use of multiplex LA-ICP-MS bioimaging of brain tissue from a Parkinsonian mouse model [23]. They stain brain tissue with metal-coded antibodies. In detail, Neumann et al. [23] employ brain tissue from the Parkinson’s disease model “line 62” and from control mice. They incubate these tissues with four antibodies relevant to the disease, and standardize the measurements to three house-keeping proteins. In addition, a new standardization approach is developed and the results compared. This new approach consists of coating specimens with gelatin, and printing an indium-doped ink with a commercial ink jet printer. This is considered as breakthrough for quantification in LA-ICP-MS bioimaging. Contrarily to normalization using house-keeping proteins, which still leads to high background signals even at high-resolution, the normalization with indium-doped ink improves the signal-to-noise ratio, even when small laser spot sizes are used, and it further improves by overlaying tissue specimen with gelatin. The authors find that “Line 62” mice have more α-Synuclein and gliosis, but decreased numbers of neurons, as also confirmed by conventional immunohistochemistry methods. However, differences between “Line 62” and controls for tyrosine hydroxylase are only detected by LA-ICP-MS. The herein presented examples show that LA-ICP-MS offers perfect insights into metal and metal drug distribution, and is invaluable for research into new cancer drugs, or as an aid to diagnosis, or the assessment of the effectiveness of metal anti-cancer drugs during treatment. Particularly, the combination of metal-tagged antibodies and LA-ICP-MS allows for precise targeting and detection, with the highest sensitivity.

The conjugation of antibodies with metals as a prerequisite for subsequent LA-ICP-MS determination leads perfectly to the next technique presented, namely, elemental tagging. 

## 4. Elemental Tagging in Inorganic Mass Spectrometric Bioanalysis

As already stated above, inductively coupled plasma-mass spectrometry (ICP-MS) has become one of the most versatile and sensitive tools in bioinorganic analytical chemistry for the determination of most of the elements present in the periodic table. Besides their importance for manifold biochemical reactions, cellular signaling, and the maintenance of chemical or redox equilibria in living systems, chemical elements may also be used as natural tags in a wide range of biomolecules. This opens the unique possibility to detect and quantify these biomolecules with outstanding sensitivity and selectivity, either via their natural elemental tag, or via a conjugated or artificial labelled element tag [27,28].

Natural elemental tags: The need for a quantitative proteome profiling strategy led to a variety of approaches for the stable isotope tagging of proteins. Typical natural tags are selenium atoms as essential parts of selenoenzymes, Se-carrying proteins, or Se metabolites such as selenocysteine or selenomethionine. Numerous Se speciation papers have been published and partly summarized. As an example, a comprehensive review of the current understanding of the metabolism of common dietary selenium compounds—selenite, selenomethionine, methylselenocysteine, and selenocystine—including a discussion of the evidence for the various metabolic pathways and their products was published by Weekley et al. [29]. The authors link antioxidant, pro-oxidant, and other mechanisms of the dietary selenium compounds to their disease prevention and treatment properties. The evidence of the impact of different Se species on these mechanisms is evaluated, with emphasis on the selenium metabolites involved in vitro, in cells, and in vivo, which motivated the authors to consider that dietary selenium compounds as prodrugs. Their analytical procedure relies on separating the Se compounds from each other (e.g., by HPLC), followed by ICP-MS detection via their natural elemental tag, the selenium atom(s) in the respective molecules. Further use is made of selenium as a natural elemental tag when elucidating seleno-compounds before (in serum), or behind (in cerebrospinal fluid), the neural barriers in neurodegeneration research projects [30,31,32,33]. This is of importance, since brain and cerebrospinal fluid (CSF) levels of Se species are independent of respective blood levels [34,35]. Advances in chemical selenium speciation [36,37,38,39], and the introduction of genomic, autoradiographic, and proteomic techniques [40], provide new insights in the studies of brain selenium biochemistry and neurotoxicology, which are comprehensively summarized in [38]. The need for a detailed consideration of the selenium species, using the selenium as natural elemental tag in speciation analysis, is also outlined in [41]. While, so far, observational studies yield inconsistent results when addressing the relationship between total selenium (but not selenium species) and human health, contrarily, randomized trials show a fairly consistent pattern, suggesting null or adverse effects of the total selenium. In turn, Vinceti et al. highlight the potential for exposure misclassification of observational epidemiologic investigations, based on only total selenium content in blood and possibly other tissues. These authors analyze selenium and, additionally, selenium compounds in serum of fifty subjects of the population of the Modena municipality, by using selenium as elemental tag for selective compound analysis (selenium speciation), and comparison to total selenium measurement. 

Arsenic is another metalloid that can be considered as natural tag in As compounds. Today, As species are regularly monitored in food-safety investigations, and in urine samples of occupational monitoring settings, to quantitatively differentiate exposure to toxic arsenic compounds (e.g., inorganic As(III) or As(V)) from non-toxic ones (e.g., arsenobetaine). This analytical approach is commonly known as arsenic speciation [42]. Lucio et al. employ this method to investigate the association of As exposure, measured as total arsenic and arsenic speciation in urine and blood, with the prevalence of diabetes mellitus type 2 (DMT2) in a representative sample of Croatian adults. Significant differences for total As and As species in urine of participants from eastern and western Croatian regions are observed. Inorganic As species, the prevalent species in drinking water, is mostly found in urine of patients from eastern Croatia, whereas arsenobetaine (an As metabolite present in seafood) is the most abundant As species in the urine of participants from the western (coastal) region. Total As and As species in urine positively correlate with glycosylated hemoglobin A1c (HbA1c) values and negatively with albuminuria. Obtained results show no clear evidence for an association between exposure to As through drinking water and the prevalence of DMT2 [43]. Although arsenobetaine is frequently ingested by humans and rapidly excreted unchanged in urine, little is known about its transport in the human bloodstream. An assessment about whether this transport involves binding to plasma proteins was performed by Pei & Gailer [44], using SEC-ICP-optical emission spectrometry (OES). ICP-OES is an elegant alternative to ICP-MS detection, as long as the target elements concentrations are not in an ultra-trace range and, thus, are above the limit of detection (LOD) of the hyphenated HPLC–ICP-OES speciation system. The particular advantage is that the selected element emission lines can be simply monitored simultaneously at their respective “clear” (non-interfered) element emission lines, without sophisticated interference elimination. Simultaneous monitoring of As, Cu, Fe, and Zn in the SEC column effluent allows for arsenobetaine determination relative to that of the major Cu-, Fe-, and Zn-containing metalloproteins. Their results indicate that arsenobetaine does not bind to plasma proteins, and that SEC–ICP-OES is a rapid, and comparatively cheaper, tool than SEC–ICP-MS for probing toxicologically and pharmacologically relevant interactions between organometalloid compounds and human blood plasma constituents.

Aside from trace metals and metalloids, heteroelements, such as phosphorus or sulfur, also represent naturally occurring, powerful elemental tags. The determination of phosphorus was, therefore, used recently for the sensitive and selective detection and quantification of dsDNA, styrene oxide, and melphalan DNA adducts [45,46,47]; the last of above-mentioned needs exceptional sensitivity. Styrene oxide is a potential carcinogen in humans, generating DNA adducts and potentially causing changes in DNA structure. Thus, sensitive monitoring and quantification of such DNA structure changes is of paramount importance. Element-specific detection with ICP-MS and phosphorus as the elemental tag in nucleotides is an elegant measure for quantification. The available modern ICP-MS technologies, based on high spectral resolution or dynamic reaction cell (DRC) mode, combined with multi-quad techniques, provide the necessary powerful detection capability to overcome or resolve spectral interferences on the ^31^P isotope, while highly efficient separation techniques installed in front of ICP-MS are functioning as a sample delivery system. They enable the separate, time-delayed measurement of the different nucleotide adducts. Such an ICP-MS-based, element-specific detection of phosphorus from HPLC-separated nucleotides ensures structure-independent sensitivity and detector response. Quantification can then be simply related to the ^31^P-calibration, and the knowledge about P-stoichiometry per nucleotide. 

Sulfur represents the most versatile naturally occurring elemental tag detectable by ICP-MS that can be found in biomolecules. It might, therefore, be possible to determine the majority of proteins on the basis of the known stoichiometry of their sulfur tag, thus, allowing absolute protein quantification [48]. In this context, the modern element speciation techniques provide the possibility of parallel monitoring of the elements’ concentration of interest (e.g., Cu or Zn as the key elements in enzymes) in specific HPLC-separated element peaks, together with the S concentration in those chromatographic peaks. For protein quantification, the knowledge of the tag (S) stoichiometry is essential [49]. The combined knowledge of the sulfur concentration and S stoichiometry of the protein allows calculation of the element saturation (Cu, Zn) of the respective protein. Species-specific isotope dilution analysis might offer an attractive alternative for their quantitation with respect to analytical quality control [27]. For quantification using sulfur isotopes, the application of a species-unspecific isotope dilution method, based on enriched sulfur isotopes, has the potential to quantify all proteins of interest with known sulfur stoichiometry, according to the compound-independent elemental response in ICP-MS [28,50]. It is noteworthy to mention that monitoring sulfur as a tag for proteins is simpler when established with ICP-OES as chromatographic HPLC detector, since its concentration is typically rather high—ICP-MS detection likely reaches overflow intensities—and emission lines are free from interferences. However, substantial limitations can be the low concentrations of some trace elements in body fluids, which in turn may represent the elemental tags of metalloproteins of interest. Contrary to ICP-MS, a parallel monitoring might be hindered due to metal-tag concentrations being below the LOD of the HPLC–ICP-OES system.

Artificial elemental tags: An advanced approach in this field is the artificial labelling of proteins of interest with an elemental tag by means of derivatization. Tagging is the fastest and easiest way for protein detection and quantification. However, reaction chemistry for the labelling of proteins, via their primary amino groups or cysteine residues, is still problematic. Metal nanoparticles (NPs), such as AuNPs [51,52] were investigated as labels of biomolecules for detection by ICP-MS, and investigations were further extended to NPs of elements, with particular sensitivity in ICP-MS detection, i.e., NaEuF, NaTbF, and NaHoF [53]. Notably, a considerable part of such chelates and polymers containing NP-types does not contribute to the amplification of the analytical signal. Therefore, an advantage of pure metal NPs, similar to nanoclusters, is that they provide a higher ratio of the detected metal atoms per label size. Cruz-Alonso et al. use this technology in an ophthalmological research project: Metallothionein (Mt) is bioconjugated with gold nanoclusters as specific tags, and the conjugates used for bioimaging Mt in ocular tissue, using LA-ICP-MS. The study is extended to quantitative imaging of specific proteins in retina by LA-ICP-MS, after bioconjugation with gold nanoclusters [26,54,55]. Recently, Menero-Valdes et al. successfully used iridium nanoclusters (IrNCs) as highly sensitive and tunable elemental labels for antibodies (Ab) in glaucoma research [56]. Glaucoma is a complex group of neurodegenerative eye disorders, of multifactorial origin, characterized by optic nerve damage, retinal ganglion cell death, and irreversible loss of visual field [57]. Primary open angle glaucoma (POAG) causes excessive production of aqueous humor, or outflow system obstruction. Dysfunction in the aqueous humor outflow results in the elevation of the intraocular pressure [58]. In Menero-Valdes et al.’s study, the applied cluster methodology, even with 250 Ir atoms per cluster, shows lower limits of detection than those previously reported with other metal nanocluster labels. Compared to current methods for the quantification of apolipoprotein E (APOE) and immunoglobulin E (IgE), the Ab/IrNCs methodology used here provides reduced heterogeneity of results. As an advantage, the amplification factor, and as such, the detection limits (down to tens of fg protein/µL) could be tuned by a suitable selection of the Ab:IrNCs molar ratio. The authors apply their newly developed technique to the determination of APOE and IgE in a small sample of human aqueous humor. They find insignificant higher APOE levels in aqueous humor of POAG patients (807 ± 180 ng/mL) compared to control subjects (774 ± 361 ng/mL), and also higher values than in previous literature (500 ng mL) [57]. IgE levels are lower compared to previous references [59,60,61].

In elemental tagging set-ups, several technical ensembles can be found, such as combining SDS-PAGE protein separation instead of HPLC or CE separation, natural or artificial elemental tagging, and LA-ICP-MS for screening the separated protein bands on the gel. Sodium dodecylsulfate polyacrylamide gel electrophoresis (SDS-PAGE) combined with LA-ICP-MS is used by Neilsen et al. [62], being, at that time, a new sample introduction technique for metal detection in PAGE separated proteins. Marshall et al. [63] and Wind et al. [64] analyze phosphorylated proteins separated by SDS-PAGE after blotting onto membranes, and Becker et al. apply laser ablation for parallel, direct detection of metals, sulfur, and phosphorus in human brain proteins [65,66]. For deeper insight about the ensemble of PAGE, labelling, and LA-ICP-MS, the reader is directed to comprehensive reviews in literature, such as [66,67].

As shown above, tagging is an effective way to either quantify biomolecules, such as proteins or nucleotides, using natural tags (As, P, S, Se…) with outstanding sensitivity and selectivity, or for metal-containing drugs (using the metal as tag) during drug development, and particularly for diagnosis and therapy control. The use of metal-tagged antibodies finally allows precise molecule targeting and detection with advanced sensitivity and selectivity.

## 5. Iron Redox Speciation

Within health research, it is evident that dyshomeostasis in iron metabolism is a hallmark in the pathophysiology of multiple diseases, resulting in oxidative stress (OS) and, ultimately, triggers a programmed, necrotic, iron-dependent cell death, known as ferroptosis (FPT). This is particularly true for neurodegenerative brain disorders, such as Alzheimer’s and Parkinson’s disease [9,68,69]. A key feature in the iron-mediated progression of OS and FPT is the state of the redox-couple Fe(II)/Fe(III). While Fe(III) is redox-inactive, Fe(II) potently generates reactive oxygen species (ROS) which can lead to membrane lipid peroxidation via Haber–Weiss and Fenton reactions [10,70]. On a cellular level, excess ROS and peroxidized phospholipids hamper the integrity of proteins, lipids, and DNA [71,72]. Therefore, quantitative Fe(II)/(III) redox speciation analysis is of paramount importance in research of the relationships between disease pathologies and OS. In 2014, Fernsebner et al. introduced a ion chromatography ((IC)-ICP-OES)-based method for analyzing the Fe(II)/Fe(III) ratio in rat brain extracts after chronic oral manganese exposure [73]. They aimed to decipher the ongoing neurodegenerative mechanisms in manganese-affected brains, since chronic occupational or environmental Mn exposure is prone to cause degeneration of dopaminergic neurons, inducing a Parkinson-like complaint called manganism. These authors use cation-exchange chromatography for separation of Fe(III) from Fe(II), and ICP-OES as the element-specific chromatography detector. ICP-OES is best suited for the high iron concentration in rat brains, and the simple parallel measurement of sulfur chromatograms for protein determination, along with Fe(II) and Fe(III) monitoring. They observe a significant (*p* = 0.015), 3-fold increase of the Fe(II)/(III)-ratio in the test rat brain extracts after 5 months low dose (oral) Mn-exposure, compared to control rat brains. This increase correlates positively in brain extracts with markers of oxidative stress such as glutathione disulfide (GSSG), prostaglandins, and 15-hydroxyeicosatetraenoic acid, a major arachidonic acid metabolite from the 15-lipoxygenase pathway and marker of lipid peroxidation, each determined with electrospray ionization Fourier transform resonance mass spectrometry (ESI-FT-ICR-MS), but correlates negatively with the amyloid precursor protein (APP) concentration [74]. Fernsebner’s investigation implies that even after low-dose oral manganese exposure, iron-mediated oxidative stress and lipid peroxidation is triggered. Consequently, to analyze probable similar outcomes in humans, Fernsebner’s method is further refined and adopted for redox elements Cu, Mn, and Fe, as well as S, for protein monitoring to the respectively low element concentrations in human cerebrospinal fluid (CSF) by [17]. This improved method uses cation-exchange chromatography (CEC) coupled to ICP-sectorfield-mass spectrometry (ICP-sf-MS), and provides simultaneous redox speciation of iron (II/III), manganese (II/III), and copper (I/II) in CSF, at detection limits each below 1 µg/L. The redox active metals copper, manganese, and iron play an important role in protein aggregation [10], via the generation of free radicals and, consequently, their redox state needs to be monitored in neurodegenerative conditions. By testing 38 CSF samples, the authors prove the satisfactory sensitivity and precision of their method, and recommend the technique for primary epidemiological and neurobiological studies. However, a drawback of the so-called, LC-based speciation methods is a relatively low sample throughput, partly caused by sample matrix sticking to stationary phases of chromatographic columns, followed by prolonged purging cycles, and a slow decrease in analytical precision. Therefore, the group around Michalke extend their redox speciation capabilities to capillary electrophoresis (CE)-ICP-DRC-MS methodology [75]. An additional benefit of this approach is the capability to gain information about the iron load of ferritin, aside from Fe(II) and Fe(III) concentration. Sample analysis time is shortened to 4 min per sample, with detection limits of 3 µg/L iron per Fe species, which is still acceptable for CSF measurements. The applicability of the method is first shown by analyzing CSF samples from neurodegeneration research. CSF samples from patients considered neurologically healthy after a clinical chemistry check of their CSF, reveals an iron concentration of 11.9 ± 3.4 µg/L and a Fe(II)/Fe(III) ratio of 0.68, the latter being in a regular range not pointing to OS. For even deeper insight into manganese-induced derangement of iron metabolism, a set of different analytical methods, including CE-ICP-DRC-MS, was subsequently employed to analyze extracts of neuronal SH-SY5Y cells after Mn exposure [74]. From the data, it is hypothesized that the Mn-altered IRP1/IRE-binding affinity coordinates the translational suppression of neuroprotective APP and H-Ferritin. In turn, that leads to a profound increase in redox-active iron (Fe(II)), providing a more complete explanation of the Mn-induced shift in the Fe(II)/Fe(III) ratio and neurotoxic oxidative stress accumulation. However, iron-related oxidative stress is not limited to neurodegeneration. In cancer-related research, the CE-ICP-DRC-MS technique proved useful, too. The comprehensive paper of Vara-Perez et al. reports about BNIP3-induced promotion of HIF-1α-driven melanoma growth, by confining iron homeostasis, as expressed by a drastic increase in the Fe(II)/Fe(III) ratio. This increase in the Fe(II)/Fe(III) ratio is a direct result of the elevation of ROS-promoting ferrous Fe(II), while—notably—the total iron concentration remains constant. Both the increase in Fe(II), together with constant total Fe, underlines the necessity to monitor the concentrations of the redox couple Fe(II)/Fe(III), instead of total iron, when investigating molecular mechanisms in OS-related pathologies, as shown in subsequent Figure 3. In this paper, BNIP3-melanoma cells display increased intracellular Fe(II) levels, caused by an elevated, NCOA4-mediated ferritinophagy, the latter fostering PHD2-mediated HIF-1α destabilization [76].

Another clinical application is shown by Kruszewska, Sikorski, et al. (2020), when they monitor superparamagnetic iron oxide nanoparticles (SPIONs) [77]. SPIONs attracted much attention, due to their medicinally attractive properties and their possible application in cancer diagnosis and therapy. The range of possible biomedical applications of SPIONs is wide, including magnetic resonance imaging, drug and gene delivery, magnetic hyperthermia therapy, photodynamic therapy, phototherapy, and chemotherapy [78,79]. They can be applied as theranostic nanoprobes, having both therapeutic and diagnostic properties. To overcome the lack of appropriate methods, which enable the quantitative monitoring of the particle changes in a physiological environment, Kruszewska et al. introduce a CE-ICP-MS/MS method. They optimize the type and flowrate of the collision/reaction gas, aiming for simultaneous Fe and S monitoring. Together with the short analysis time and low sample consumption, their technique enables the separation of differently charged SPIONs by CE, and their quantification by tandem ICP-MS in relevant samples, for cancer diagnostics and therapy.

Iron redox speciation analysis has now matured to a method with reasonable sample throughput, and which provides a valuable measure for OS condition by quantitative determination of the Fe(II)/Fe(III) ratio. The method was already applied to human samples for diagnosis support (CSF, liver biopsy in hemochromatosis), and in cellular extracts of animal experiments in the frame of basic research of disease pathogenesis.

## 6. Copper Speciation in the Frame of Cu-Related Diseases

The knowledge of copper (Cu) distribution in blood contributes to a better understanding of copper metabolism and of related diseases, such as Wilson’s disease (WD), one of the most severe Cu-related disorders [80]. WD is a genetic disorder of copper metabolism, which is characterized by Cu accumulation in various body tissues, mostly the liver, brain, and the cornea of the eye (Kayser–Fleischer ring). Mutations in the Cu-transporter gene “ATP7B” cause this problematic Cu overload, since its participation in biliary Cu excretion and Cu incorporation into apo-ceruloplasmin (apo-CP), for the synthesis of enzymatically active holo-CP, is impaired [81]. In WD, this impaired Cu metabolism leads to low levels of holo-CP released into the blood, which in turn results in high levels of so-called exchangeable copper and/or free copper. CP is the major copper-carrying protein in blood, and accounts for 85–95% of serum copper. Cu is tightly bound to CP and is non-toxic. Cu-containing holo-CP oxidizes Fe(II) to Fe(III), maintaining the iron redox balance and lowering the risk for oxidative stress [82,83]. The remaining circulating Cu is either, to a very small extent, free Cu(II), or the so-called exchangeable Cu fraction (Cu–Ex). Cu–Ex is loosely bound, its main part complexed with albumin, and, to a lesser extent, α-2-macroglobulin, or to low-molecular-weight compounds such as histidine [84,85,86]. Exchangeable and free Cu levels in the liver and brain, when in excess due to the “ATP7B” mutation, become toxic to cells, and cause cell damage and cell death. In turn, high amounts of exchangeable and free Cu levels are the consequence of low holo-CP levels, and the associated reduced oxidation of Fe(II) to Fe(III), i.e., accumulation of Fe(II) and OS.

Consequently, there is considerable interest in the amount of extractable and free Cu in serum as a concept for clinical purposes in the treatment and diagnosis of WD patients [87,88,89]. This hazardous Cu fraction was historically calculated from CP concentration, analyzed by biochemical assays, which were subtracted from spectroscopically determined total Cu values in serum. Although this approach is still established in clinical practice, it is prone to unreliable values, which originate from the limited accuracy of the nephelometry measurement of CP [90,91,92]. The CP antibody used is not specific for holo-CP, but also takes apo-CP, thus, overestimating the Cu–CP fraction [92]. Therefore, novel, and successively more precise, methods were developed in recent years. ElBahti et al. [80] introduce a simple and cheap method for determining the exchangeable copper using ultra-filtration (UF) and electrothermal atomic absorption spectrometry (ETAAS) for Cu determination in UF fractions after complexation with EDTA. From the UF devices, contamination problems are identified, as well as stability problems of Cu species after sampling and storage. A loss of 40% and 60% of Cu in UF is observed by these authors when plasma is kept for more than 31 days at +4 °C and −20 °C, respectively. This point is crucial in the sample handling procedure, and could compromise the reliability of results. Deep-freezing the plasma at −40 °C, immediately after sampling, offers an alternative. Notably, this instability issue is a general problem in Cu speciation, affecting the reliability of results, independent of the subsequent determination procedure. The equilibration in vitro is incomplete and the proteins, such as macroglobulins present in plasma, seem to capture Cu atoms from either exchangeable or free Cu, but, inversely, also from the gradually degrading Cu–CP complex [93,94]. Similar observations are also made in [18], who report about Cu leaching even in tightly bound Cu from CP, and who recommend that the term “exchangeable Cu” should always be connected to the operational conditions of the respective experiment, since those conditions can significantly influence the amount of this fraction. Summarizing, these uncertainties are the reason why the percentage of copper bound to ceruloplasmin is still a matter of debate. It is usually reported to fluctuate between 90 and 95%, but recent studies using size-exclusion chromatography coupled to inductively coupled-mass spectrometry (SEC-ICP-MS) claim that this percentage is rather at 70% [95,96,97]. According to these studies, in healthy subjects, 15–19% of the copper is bound to albumin, 7–15% to transcuprein, 65–71% to ceruloplasmin, and less than 2–5% of copper remains free and/or bound to amino acids. Other papers, too, use chromatographic methods with hyphenated ICP-MS detection as an alternative to operationally defined UF fractionation. In 2000, Inagaki et al. published a method with a 5 h offline sample pre-treatment followed from SEC-ICP-MS [98]. However, such long pre-analytical and analytical times per sample are absolutely unsuitable for high-throughput or clinical practice. Muniz et al. present an anion-exchange chromatography (AEC)-ICP-MS method for detection of Cu bound to ceruloplasmin, which is already much faster but still takes more than 22 min per sample, and only differentiates between Cu bound to albumin, versus bound to ceruloplasmin, aside from total Cu [99]. The recently published method from Solovyev et al. represents a significant improvement in terms of analysis time and Cu species discrimination. With their AEC-ICP-MS/MS method, they determine free Cu, Cu–human serum albumin (HAS), and Cu–CP in serum samples within 8 min [18]. The use of DRC technology, with triple-quad ICP-MS, provides the parallel measurement of copper and the important biological hetero-elements S and P, by effective removal of polyatomic interferences, even in real biomedical samples at physiological concentrations. Solovyev et al. test their AEC-ICP-MS/MS method in a small sample of blood sera from WD patients and healthy controls, but the method provides a relatively simple copper quantification, with the limit of detection of 0.1 µgL^−1^ (limit of quantification 0.4 µg/L). In 2020, Quarles et al. also refer to the lack of established direct determination methods of extractable copper, and particularly admonish reliable very quick methods [100]. Therefore, they develop a combination of an automated sample preparation and LC system with ICP-MS detection to differentiate two copper fractions, “bound Cu”, and extractable Cu in serum. The “bound Cu” is the sum-peak of Cu–CP and (if present) Cu–EDTA, whereas the extractable Cu is the sum of Cu–HSA, Cu low-molecular-weight compounds (LMW), and free Cu^2+^. The main aim of this work is to develop a reliable method for quantifying extractable copper that is as fast as possible, accepting a loss in species differentiation exceeding the fractions “bound” and “exchangeable”. This LC-ICP-MS method takes only 250 s for sample preparation and analysis, and a subsequent column recondition step resulting in a total sample-to-sample time of 6 min. Method validation is performed by measuring serum collected from either control (Atp7b+/+) or Wilsons disease rats (Atp7b−/−). The extractable Cu is 4.0 ± 2.3 µM Cu in healthy control rats, but 2.1 ± 0.6 µM Cu in healthy Wilson rats and 27 ± 16 µM Cu in diseased Wilson rats, respectively. The ratio of extractable Cu/bound Cu is 6.4 ± 3.5%, 38 ± 29%, and 34 ± 22%, respectively. According to the short analysis time and proven reliability, this method may be of diagnostic value in clinical practice for Wilson’s disease. Most recently, del Castillo Busto introduced another AEC-ICP-MS method for copper speciation with determination of exchangeable copper being relevant to Wilson’s disease. The specialty of this approach is the use of isotope dilution mass spectrometry as means for quality control and the high quantitative accuracy [101]. The methodology is characterized in terms of selectivity, sensitivity, precision, and robustness. Three human sera from commercial control materials are successfully analyzed and prove the method can discriminate between healthy and WD populations, in terms of Cu–albumin content. The methodology with quantification specifically based on isotope dilution (ID)-MS developed in this work promises to be invaluable for quality control assessment, and WD drug monitoring.

Specifically, the recently published methods using AEC-ICP-MS or CEC-ICP-MS [18,100,101] provide fast analysis time per sample, good accuracy, and acceptable or very good discrimination between the relevant Cu species, and are ready for routine use in clinical practice.

## 7. Single Cell ICP-MS (SC-ICP-MS)

The analysis of single cells is a growing research field in many disciplines, including toxicology, medical diagnosis, drug and cancer research, and metallomics. Numerous analytical ICP-MS-based methods were developed or improved, to allow the analysis of individual cells and their cellular compartments [102,103,104]. Analysis of individual cells provides highly valuable information on cell-to-cell variance, especially within an isogenic cell population. Cells display significant heterogeneity in size, individual physiological status, or access to nutrients, and specifically drugs, which can lead to varying metal uptake rates and metal concentrations, which is important in metal drug research and their clinical application [105]. To understand the biochemical status of the entire cell population under consideration, the biology of individual cells needs to be studied [106]. In element-related single cell analysis, two different concepts of sample introduction to ICP-MS are mainly applied today. The most common concept of single cell ICP-MS (SC-ICP-MS) is based on the nebulization of a cell suspension, with direct rapid analysis of individual cells for their metal content. Using pneumatic nebulization for sample introduction into the ICP-MS is often used to directly measure cell parameters (especially biomarkers) in a large number of cells in a very short time (1000 or more cell events in 1 s). Contrary to conventional ICP-MS, which looks at a continuous signal, SP-ICP-MS looks at discrete signals, where a cell suspension is nebulized through a dedicated introduction system, managing each cell entering the plasma one after the other. Maintaining the critical balance between efficient separation of the cells into individual cells (no clustering) through well-controlled sample buffer conditions on the one hand, and avoiding cell bursting on the other hand, is an extremely important prerequisite during sample preparation and cell nebulization for SC-ICP-MS. Cell burst results in increased background continuous element signal (such as in conventional ICP-MS), and the aimed discrete signals per cell are lost. For minimizing coincidence of multiple cells entering the plasma, the cell concentration should be around 100,000 cells/mL. In the ICP-plasma, each single cell is ionized and the resulting ion burst from the intrinsic metal is detected by the ICP-MS as a sharp, very short signal spike [107]. In this optimal situation, one cell yields one ion burst, with the intensity of the resulting signal being related to the size of a cell (nm), and the number of pulses being related to the cell concentration (part/mL). Thus, on the instrumental side, for successful SP-ICP-MS, a rapid, continuous measurement is mandatory, having the ability to monitor and quantify discrete, very short pulses of positively charged ions in a time-resolved manner, using microsecond data acquisition rates [107]. Such short dwell time not only ensures that the spike signals relate just to one cell, but additionally achieves a higher signal-to-noise ratio [108]. The first successful application of this approach of direct-label-free analysis of single cells by ICP-MS was reported by Houk’s group in 2005 [109]. They measured the signal of uranium incorporated intrinsically in Bacillus subtilis, using a micro-concentric nebulizer. At the beginning of SC-ICP-MS, the main research focus was about suitable systems for the introduction of intact single cells into ICP-MS. As an example, Verboket et al. present a microfluidic system for droplet generation, and direct injection of cell suspensions into the ICP-MS as a technical improvement for sample introduction [110]. In the same year, the authors around Bodenmiller introduced a sophisticated method for increasing detection sensitivity and specificity for particular cellular compounds, through the use of antibodies tagged with rare metals. Multi-element scanning in respectively treated cells allows for the acquiring of high-dimensional parameters of cell components tagged by those antibodies/metals [111,112].

Single cells are currently also standard models to understand nano–bio interactions. To elucidate how the physical–chemical properties of a nanomaterial are in fact connected with the local distribution and the number of nanoparticles, the uptake of a nanomaterial has to be quantified, and the intracellular fate, including major sites of accumulation, determined. The application of an elemental barcode by Bodenmiller et al. increases the high-throughput cell screening by mass cytometry. To achieve this, seven maleimide-DOTA loaded with lanthanide isotopes were used to generate 128 combinations, enough to barcode each sample in a 96-well plate [113]. The cells in each well were labeled with a unique isotope composition, and then they were pooled into a single tube for the immunoassay with metal-labeled antibodies. For each of 27 inhibitors, 14 phosphorylation sites were analyzed in 14 PBMC types at 96 conditions, resulting in 18,816 quantified phosphorylation levels from each multiplexed sample. For more details about data handling of massively, multiparametric single-cell assays analyzed, the reader is recommended to read the review from Zivanovics et al. [114]. SC-ICP-MS has also been applied to investigate cellular bioavailability of arsenite. Meyer et al. incubate cells with inorganic AsIII [108]. They apply a specific sample and cell preparation method, which first detaches the cells from culture dishes by trypsinization. Afterwards, they wash the cells with ice-cold PBS buffer, count and pelletize them and subsequently, they are re-suspended in ice-cold, deionized water before being directly analyzed by SC-ICP-MS/MS. In contrast to fixation procedures typically used elsewhere, this unsophisticated procedure avoids cell membrane permeabilization, thereby minimizing the loss of cytosolic metal species. The authors discuss that the presented method is less time-consuming and avoids the loss of cytosolic metal species, which is normally caused by cell fixation. Information on metal distribution among the cell population is still available, and gives further insights into uptake, accumulation, and release on the single-cell level. Short dwell times of few milliseconds provide suitable time resolutions for the calculation of single-cell metals [108]. This study demonstrates that SC-ICP-MS is a solid alternative to the commonly used digestion protocols for the assessment of cellular arsenic bioavailability. In several recent studies, SC-ICP-MS is increasingly involved in cellular Pt-drug penetration studies for anti-cancer drug research. Resistance to cisplatin is connected to three molecular mechanisms: increased DNA repair, altered cellular accumulation, and increased drug inactivation, e.g., by drug–protein bonds. In this context, SC-ICP-MS can be of inestimable value as a diagnostic tool in precision medicine. Clinicians are able to make informed decisions on a patient’s cisplatin resistance status, and the consequent assessment of tumor response to Pt drugs allows them to tailor the chemotherapy program based on the biology of the disease [115]. As each cell is unique and is present in its own growth stage, as well as the cell population having a distribution of different cell volumes, the use of SC-ICP-MS is particularly useful for the quantification of cisPt in single cancer cells. Knowing this, Gale et al. reach out even beyond single cells when targeting their analysis to Pt-metallodrugs in cellular sub-structures [116]. As a prerequisite, the implementation and advanced development of single SC-ICP-MS supports these authors in anti-cancer research, when they analyzed not only single cells, but even isolated nuclei. The cisPt uptake is compared between a wild type (wt) cancer cell line and related resistant sublines. Nuclei are isolated from cells treated with 30 μM cisPt, and the purity of the nuclear extract assessed by Western blots, verifying the absence of endoplasmic reticulum and cytosolic contaminants. A key finding is that the amount of internalized cisPt is lower in resistant cell lines and their nuclei compared to wt cells. This might be an explanation for the reduced impact of cisPt in cells, which appear to be resistant to this anti-cancer drug [116]. Aside from platinum-based anti-tumor drugs, the search for, and development of, further metallodrugs is ongoing, and SC-ICP-MS has also found its way into those research settings. In the study of Zheng et al., the uptake of a newly developed nanomaterial anti-tumor Gd@C_82_(OH)_22_ is compared to cisplatin, regarding concentrations in HeLa and 16HBE cells, at different exposure times and doses [117]. Gd@C82(OH)22 exhibits higher bioavailability and lower toxicity than cisplatin in vitro, which opens up the chance to inhibit tumors at a lower dose. The herein used SC-ICP-MS approach was validated against regular ICP-MS in the digests of the cells. The SC-ICP-MS method determines the metal drugs in single cells, and helps to better understand the uptake of the drugs due to cell variation and basic biological processes at a single cell level.

The second, very successful sample introduction concept in single cell ICP-MS is based on LA-ICP-MS, after plotting single cells on a flat surface [118]. This type of sample introduction is often used for the imaging, or immuno-imaging, of single cells. Spatial resolution is a limiting factor in the beginning of such analytical approaches, but today, methods are developed with high sensitivity and spatial–temporal resolution, which allows qualitative and quantitative analysis at single-cell and subcellular levels, targeting for organelles [118]. For the latter, element-tagged antibodies are of especially high interest, and provide precise localization of the molecule of interest, combined with maximum detection power (refer also to Section 3 of this article). Jakubowski and co-workers employ this principle, and test a novel automated single spot LA-ICP-TOFMS, using a single-cell piezo-acoustic microarrayer to optically detect individual cells, generate droplets containing single cells, and dispense them onto an array [119]. This novel procedure is performed on THP-1 cells (a human leukemia cell type), with subsequent LA-ICP-MS analysis to detect naturally occurring isotopes as fingerprints of the individual cells. The technique is found to be independent of cell size, which is different to conventional nebulization ICP-MS. The group around Jakubowski extend their investigations to elemental distribution images of NIH73T3 fibroblast cells (mouse embryonic cells) and HeLa cells, which are incubated with AuNPs and Cd-based QDs [120]. The group’s analytical approach enables high-spatial resolution (2.5 μm lateral resolution), thus, providing insights into the uptake and aggregation distribution of Au and Cd within fixed single cells. Similar to routine approaches, Zheng et al. develop a high-throughput method for LA- ICP-MS single-cell analysis, also comprising a microwell array [121]. Single cells are efficiently trapped by a polydimethylsiloxane (PDMS) microwell array. After optimization, 60% of the microwells contain one single cell, and only 4% of the microwells contain more. Since the cells in the array are regularly distributed, they are easily targeted and quantitatively analyzed by LA-ICP-MS with high-throughput. In this study by Zheng et al., AgNP uptake in single human normal bronchial epithelial cells (16HBE cells) is investigated. Notably, the results reveal that biological responses to Ag exposure varies on an individual cell basis. This suggests that great insights into the biological effects of metallic NP on a bigger scale (high-throughput) can be achieved when employing microwell arrays together with LA-ICP-MS. The studies of Van Acker et al. also evaluate the potential of LA-ICP-MS as a highly resolving imaging and single-cell analysis technique. This study aims for the localization and quantification of the membrane receptors of two human breast cancer cell lines [122]. The images of this study are compared to the established fluorescence confocal microscopy pictures, which provide similar results, thereby validating LA-ICP-MS for investigating targeting vectors in precision medicine. The severe limitations of cisplatin and its second- and third-generation derivatives, all of which have significant side effects on the brain, liver, and kidneys or acquired resistance phenomena, prompted the development of new generations of anti-cancer agents based on metals other than platinum [123,124,125,126,127]. Therefore, in anti-cancer research, not only noble metals/NPs, such as Au, Ag, or Pt are on the research agenda today, but also new synthesized compounds, such as ruthenium, osmium, or palladium complexes with varied structures. Such new platinum, ruthenium, and osmium compounds, each serving as an experimental drug, together with clinically established drugs, are investigated by LA-ICP-MS, with respect to their uptake into murine tumor tissue [128,129,130]. The drugs are heterogeneously distributed along the histological features of tumor tissue. Interestingly, metal enrichment is found particularly in loose soft tissues, and at the periphery of the tumor regions. In solid parts of the tumor, samples metal levels show lower concentrations. Since about 2000, there was growing interest in organometallic palladium compounds as potential alternative anti-cancer drugs. It is mainly their good antiproliferative, high-in vitro and ex vivo anti-cancer activity, exhibited by some palladium complexes, even toward tumors resistant to cisplatin and its derivatives, combined with different modes of action compared to Pt compounds, that are the main reasons for the increasing popularity of palladium compounds as therapeutic agents, and are the key to their growing success [131]. This motivated Niehoff et al. to incubate tumor spheroids with the Pd-tagged photosensitizer embedded in poly(lactic-co-glycolic acid) (PLGA) nanoparticles, and investigate the efficiency of nanoparticle-based drug delivery [132]. They observe an accumulation of the drug in the first cell layers of the tumor spheroid. A significantly more homogeneous distribution of the photosensitizer is achieved with nanoparticle-based drug delivery compared to tumor spheroids incubated with the dissolved photosensitizer without the nanoparticular drug delivery system. Finally, Bouzekri et al. use the newly developed approach “Imaging Mass Cytometry” (IMC), where mass cytometry is combined with LA-ICP-MS. Those authors expose cells to isotopically labeled tagged antibodies, and separate and move the cells by IMC to the laser ablation system. The resulting ions are analyzed with ICP-MS and assembled into an image, which reflects the spatial subcellular configuration of proteins in a single cell [133]. This study shows strong pairwise correlation between nuclear markers pHistone3S28, Ki-67, and p4E-BP1T37/T46 in classified mitotic cells, and a negative correlation with cell surface markers. IMC is shown to expand the number of measured parameters in single cells and, thus, enables higher-dimension analysis in the field of cell-based screening.

## 8. Conclusions

In this paper, various examples from the literature were used to show that ICP-MS, in combination with sophisticated sample delivery systems, goes far beyond the already well-known outstanding detection capability of elements in biological or clinical samples; when installed in such combined systems, it enables invaluable deep insights into pathological disease processes or intracellular distribution of active substances and, thus, helps for a better understanding of biological processes and dynamics on a molecular level.

Laser ablation, as sample introduction to ICP-MS, offers outstanding insights into metal and metal drug-distribution in tissue—even on an intracellular scale. LA-ICP-MS proved its high-value for developments in new cancer drug research, as well as being an aid for diagnosis, or for assessing the effectiveness of metal-based drugs. Particularly, the combination of metal-tagged antibodies and LA-ICP-MS allows precise targeting and detection with the highest sensitivity. It also demonstrates that elemental tagging is an effective way for the quantification of biomolecules using natural tags with high sensitivity and selectivity, which is further improved with use of metal-tagged antibodies.

Today, oxidative stress (OS) is one of the most important triggers of serious illnesses, and research results show that OS is particularly mediated via a disturbed iron redox balance. Fe(III) is redox-inactive, but Fe(II) generates reactive oxygen species (ROS), which are prone to cause membrane lipid peroxidation. Iron redox speciation analysis methods are reported in several publications, and provide quantitative determination of the Fe(II)/Fe(III) ratio in a broad range of biological samples and, thus, offer a valuable marker for the oxido-reductive status of a sample. Since ferroptosis and oxidative stress are also connected to other element pathways (e.g., selenium via glutathione peroxidase 4 (GPX4), the iron redox speciation methods lend themselves to be included in answering questions relating to the maintenance of the cellular redox balance.

Copper, another redox-active element, is the key element in severe diseases such as Wilson’s disease. The amount of exchangeable or free Cu is shown to be of immense importance, and the combination of chromatographic Cu species separation with sensitive ICP-MS detection nowadays enables very good discrimination between the relevant “dangerous” Cu species, and the unproblematic main Cu compound ceruloplasmin. In this paper, different approaches, either aiming for very fast analysis or for advanced differentiation a between free and exchangeable Cu, were presented.

Finally, it is outlined that SC-ICP-MS is of inestimable value as a diagnostic tool in precision medicine. Analysis of individual cells provides highly valuable information on cell-to-cell variance, and allows valuable, biologically relevant insights into the element content of individual cells, as well as the uptake of metallodrugs and engineered nanoparticles. With SC-ICP-MS, data oncologists can make informed decisions on a patient’s cisplatin resistance status, and an informed assessment of tumor response to Pt drugs. 

Summarizing, the paper presented powerful hyphenated ICP-MS combinations, with specific sample introduction systems, which can open up new avenues in medical clinical research, which, until now, were developed in cooperation between specialized analytical chemical laboratories and a few medical research groups. This review is intended to present these promising approaches to a broader medical research community and, ultimately, to achieve broader application in clinical routine, for the benefit of all.

## Figures and Tables

**Figure 1 ijms-23-06109-f001:**
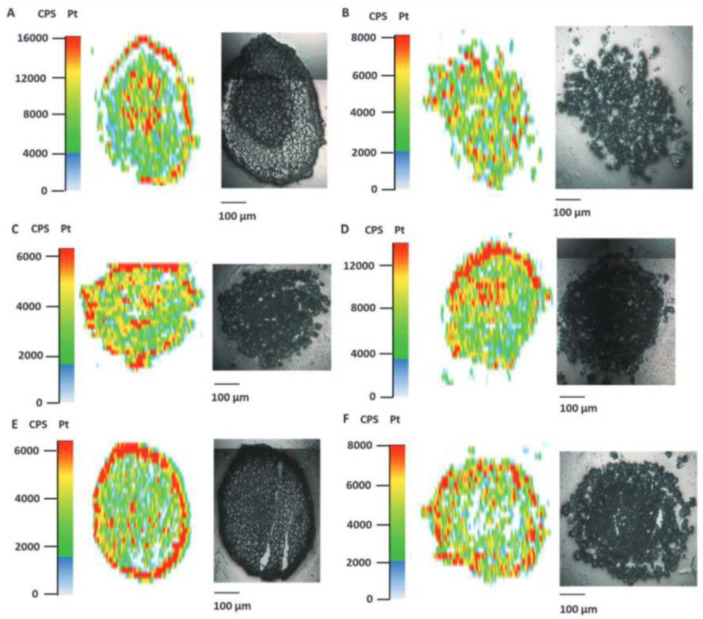
Platinum distribution in cryo-sections of HCT116 and CH1/PA-1 tumor spheroids after treatment with Pt-complexes (1–5 µM) measured by LA-ICP-MS. Reprinted with permission from Ref. [21], Copy right year: 2016, copyright owner: Oxford University Press, (**A**) HCT116 tumor spheroids treated with satraplatin, (**B**) CH1/PA1 spheroids treated with satraplatin, (**C**) HCT116 tumor spheroids treated with new “Pt-compound 1”, (**D**) CH1/PA1 spheroids treated with new “Pt-compound 1”, (**E**) HCT116 tumor spheroids treated with new “Pt-compound 2”, (**F**) CH1/PA1 spheroids treated with new “Pt-compound 2”.

**Figure 2 ijms-23-06109-f002:**
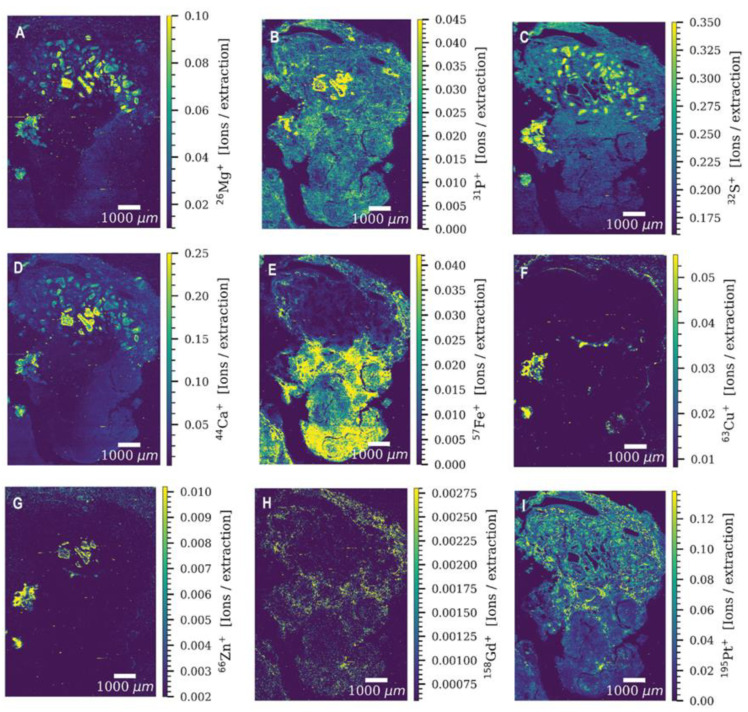
Signal intensity maps of (**A**) Mg, (**B**) P, (**C**) S, (**D**) Ca, (**E**) Fe, (**F**) Cu, (**G**) Zn, (**H**) Gd, and (**I**) Pt in a germ cell tumor tissue section of a patient obtained by LA-ICP-TOFMS imaging. The parallel line scans overlap one another by 10 µm. Reprinted with permission from [22]. Copy right year: 2020, copyright owner: Oxford University Press.

**Figure 3 ijms-23-06109-f003:**
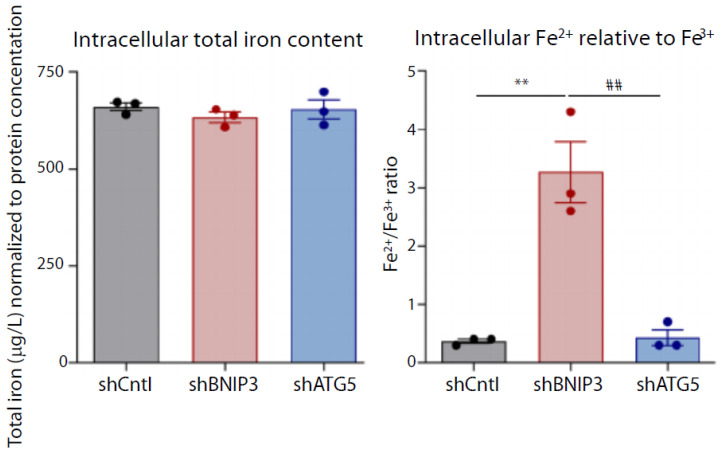
Intracellular total iron content (**left**) and Fe^2+/^Fe^3+^ ratios (**right**), measured by CE-ICP-DRC-MS in normoxic B16-F10 cells (n = 3). The data are analyzed using a one-way ANOVA with Tukey’s multiple comparisons test. ** *p* < 0.01 shBNIP3 compared against shCntl, ## *p* < 0.01 shBNIP3 compared against shATG5.Reprinted, [76]. Copy right year: 2021, copyright owner: John Wiley and Sons.

## Data Availability

Not applicable.

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
