# Peer review of "(untitled)"

_ijms, 2022, doi:10.3390/ijms23116109_

Round 1
Reviewer 1 Report
The manuscript "Review about powerful combinatios of advanced and hyphenated sample introduction techniques......" monitored a fairly large time frame even if the majority of the analyzed papers cover the last 10 years. The work as a whole is very positive and interesting. It has been developed in a critical and logical way and gives a very meaningful cross- section of the topic that the manuscript due to its very good and useful content for the study carried out can be taken into consideration to be published on Int. J. Mol. Sciences.
Author Response
Reviewer´s comment: The work as a whole is very positive and interesting. It has been developed in a critical and logical way and gives a very meaningful cross- section of the topic that the manuscript due to its very good and useful content for the study carried out can be taken into consideration to be published on Int. J. Mol. Sciences.
Response:
Thank you for evaluating the manuscript and for the positive comments.
Reviewer 2 Report
The review paper is complete and all points and sections were properly carried out and discussed. The topic is very interesting and actual. I suggest the acceptance pending revisions, particularly:
- check for typos and grammar errors
- improve the figure quality
- in the first use of acronyms please clarify the term
- please consider in case to add also PGMs (platinum group metals)
- please consider to add a section with a comparison between ICP-MS and ICP-OES
Author Response
Thank you for evaluating the manuscript and for the positive comments.
I have revised according to the reviewer´s comments:
- check for typos and grammar errors
Response: The manuscript has been cross-read and corrected from a native speaker. Typos were checked and grammar errors are now eliminated.
- improve the figure quality
Response: The figures are reproductions from reference articles. They were downloaded from the original PDFs provided by the respective publishers. The here presented figure quality is identical to the one in the original articles and cannot be improved further from this author.
- in the first use of acronyms please clarify the term
Response: First use of acronyms was checked and terms were clarified at first use.
- please consider in case to add also PGMs (platinum group metals)
Response:
Aside from already reported platinum drugs other PGM and further metal drugs are now included together with respective references [117, 124 – 133]
These considerations are at lines 682-692 and 729 – 753.
Considerations are at lines 682-692:
Aside from platinum-based anti-tumor drugs, the search for and development of further metallo-drugs is ongoing and SC-ICP-MS has found its way also into those research set-tings. In the study of Zheng et al., the uptake of a newly developed nanomaterial an-ti-tumor Gd@C82(OH)22 was compared to cisplatin regarding concentrations in HeLa and 16HBE cells at different exposure times and doses [117]. Gd@C82(OH)22 exhibited higher bioavailability and lower toxicity than cisplatin in vitro, which opens up the chance to in-hibit tumor at a lower dose. The herein used SC-ICP-MS approach had been validated against regular ICP-MS in digests of the cells. The SC-ICP-MS method was able to deter-mine the metal drugs in single cells, and helped to better understand the uptake of the drugs due to cell variation and basic biological processes at a single cell level.
Considerations are at lines 729-753:
The severe limitations of cisplatin and its second- and third-generation derivatives, all of which have significant side effects on the brain, liver and kidneys or acquired resistance phenomena have prompted the development of new generations of anticancer agents based on metals other than platinum [124-128]. Therefore, in anti-cancer research not only noble metals/NPs, such as Au, Ag or Pt are on the agenda today, but also new synthesized compounds of ruthenium, osmium or of palladium complexes with varied structures. Such new platinum-, ruthenium-, and osmium- compounds, each serving as experi-mental drug, together with clinically established drugs were investigated by LA-ICPMS with respect to their uptake into murine tumor tissue [129-131]. The drugs were hetero-genous distributed along the histological features of tumor tissue. Interestingly, metal en-richment was particularly found in loose soft tissues and at the periphery of the tumor re-gions. In solid parts of the tumor samples metal levels showed lower concentrations. Since about 2000 there has been growing interest in organometallic palladium compounds as potential alternative anticancer drugs. Mainly their good antiproliferative, high in vitro and ex vivo anticancer activity, exhibited by a couple of palladium complexes, even to-ward tumors resistant to cisplatin and its derivatives, combined with different modes of action compared to Pt-compounds are the main reasons for the increasing popularity of palladium compounds as therapeutic agents and are the key to their growing success [132]. This motivated Niehoff et al. to incubate tumor spheroids with the Pd-tagged pho-tosensitizer embedded in poly(lactic-co-glycolic acid) (PLGA) nanoparticles and investi-gate the efficiency of nanoparticle based drug delivery [133]. They observed an accumula-tion of the drug in the first cell layers of the tumor spheroid. A significantly more homo-geneous distribution of the photosensitizer was achieved with nanoparticle based drug delivery compared to tumor spheroids incubated with the dissolved photosensitizer without the nanoparticular drug delivery system.
- please consider to add a section with a comparison between ICP-MS and ICP-OES
Response:
ICP-OES has now been considered at three different positions (lines 282 – 296, lines 329 – 336, lines 405-407) in the manuscript as an alternative to ICP-MS. The respective strength and advantages or limitations in the specific applications were ruled out briefly. Respective references are included.
ICP-OES consideration at lines 282 – 296:
Although, arsenobetaine is frequently ingested by humans and rapidly excreted un-changed in urine, only little is known about its transport in the human bloodstream. An assessment about whether this transport involves binding to plasma proteins was per-formed by Pei & Gailer [A] by use of SEC-ICP- optical emission spectrometry (OES). ICP-OES is an elegant alternative to ICP-MS detection as long as the target elements con-centrations are not in an ultra-trace range and thus are above LOD of the hyphenated HPLC-ICP-OES speciation system. The particular advantage is that the selected element emission lines can be simply monitored simultaneously at their respective “clear” (non-interfered) element emission lines without sophisticated interference elimination. Simultaneous monitoring of As, Cu, Fe and Zn in the SEC-column effluent allowed arse-nobetaine determination relative to that of the major Cu, Fe and Zn-containing metallo-proteins. Their results indicated that arsenobetaine did not bind to plasma proteins and that SEC-ICP-OES was a rapid and comparatively cheaper tool than SEC-ICP-MS for probing toxicologically and pharmacologically-relevant interactions between organomet-alloid compounds and human blood plasma constituents.
ICP-OES consideration at lines 329 – 336:
It is noteworthy to mention that monitoring sulfur as tag for proteins is simpler estab-lished with ICP-OES as chromatographic HPLC detector, since its concentration is typi-cally rather high - ICP-MS detection likely reaches overflow intensities – and emission lines are free from interferences. However, substantial limitations can be the low concen-trations of some trace elements in body fluids, which in turn may represent the elemental tags of metalloproteins of interest. Contrary to ICP-MS, a parallel monitoring might be hindered due to metal-tag concentrations being below LOD of the HPLC-ICP-OES system.
ICP-OES consideration at lines 405 – 407:
ICP-OES was best suited for the high iron concentration in rat brain and the simple parallel measurement of sulfur chromatograms for protein determination along with Fe(II) and Fe(III) monitoring.
Reviewer 3 Report
The author presents an unique review summarizing the latest advances in powerful combinations of advanced sample preparation, identification and sample introduction techniques using inductively coupled plasma mass spectrometry to detect various trace elements in pathological conditions at the molecular level. This review may be of interest to a wide range of researchers working in the field of inorganic, organometallic, medicinal chemistry and pathophysiology and can be published in its present form in IJMS.
Author Response
Reviewer´s comment:
This review may be of interest to a wide range of researchers working in the field of inorganic, organometallic, medicinal chemistry and pathophysiology and can be published in its present form in IJMS.
Response:
Thank you for evaluating the manuscript and for the positive comments.
Round 2
Reviewer 2 Report
the revised version was deeply checked and all criticisms were removed. Now it can be accepted